# X-ray phase contrast reveals soft tissue and shell growth lines in mollusks

Ilian Häggmark [1] ✉, Masato Hoshino[2], Kentaro Uesugi [2] & Takenori Sasaki[1]

High-resolution 3D imaging of species with exoskeletons such as shell-bearing mollusks typically involves destructive steps. Nondestructive alternatives are desirable since samples can be rare and valuable, and destructive steps are time-consuming and may distort the tissue. Here, we show for the first time that propagation-based phase-contrast X-ray imaging can significantly increase contrast in mollusks with intact shells. By using the recently upgraded monochromator at the SPring-8 BL20B2 synchrotron beamline, we imaged six species of mollusks, showing that X-ray phase contrast enhances soft-tissue contrast. Features that are almost invisible in conventional attenuation-based micro-computed tomography (micro-CT) are clearly reproduced with phase-contrast imaging under the same scan conditions. Furthermore, this method can reveal features such as growth rings in the shell and differentiate between calcite and aragonite crystal forms. Phase-contrast imaging can thus serve as a compelling alternative when destructive methods are not an option.

[1] The University Museum, The University of Tokyo, 7-3-1 Hongo, Bunkyo-ku, Tokyo 113-0033, Japan. [2] Japan Synchrotron Radiation Research Institute (JASRI/SPring-8), 1-1-1 Kouto, Sayo, Hyogo 679-5198, Japan. ✉email: ilianh@um.u-tokyo.ac.jp

Accurately imaging the three-dimensional (3D) anatomy of a sample is indispensable in many biological disciplines. Accurate 3D imaging enables visualization of complex 3D structures and precise quantification of features[1]. However, some established imaging methods are destructive, which may not be acceptable depending on the sample used. For example, histological methods achieve excellent contrast and resolution on soft-tissue samples by using light microscopy; however, these methods require mechanical sectioning of the sample. Furthermore, the resolution is anisotropic, i.e., lower in the direction perpendicular to the sectioning plane. Artifacts can also appear due to tissue distortion in the sectioning process. This is particularly difficult for samples with hard or brittle regions, e.g., thick exoskeletons, such as shells in mollusks. An alternative method that is both nondestructive and reduces the above problems is micro-computed tomography (micro-CT). The drawback of this methods is that the soft-tissue contrast is inferior to that of histology, as the contrast mechanism relies only on X-ray attenuation. Preparation of the sample such as paraffin embedding to improve contrast is routinely used, but the treatment can also alter the sample. The problem with limited contrast is complicated further in samples with hard encasings. The area of interest in this study is malacology, the study of mollusks, but the same problems also apply to other phyla and sample types such as crustaceans[2] and the brains of vertebrates[3]. Micro-CT has been used in a few malacology studies[4–7], however, the image quality is limited because high X-ray energy is needed to achieve sufficient transmission through the shell. This lowers the amount of X-ray interactions and thus reduces the contrast, particularly in soft tissue[8–10]. In some regards, soft tissue encased in absorbing materials is a more difficult imaging problem than the presence of dense absorbing structures such as bone or teeth. There is no path for the X-rays to travel through the encased tissue that does not also pass through the encasing. Furthermore, beam hardening, i.e., artifacts due to a broad X-ray energy spectrum in combination with samples with high variations in absorption, obfuscates soft-tissue features with weak contrast. These limitations necessitate the removal of the shell to achieve sufficient soft-tissue contrast, e.g., by decalcification[10]. However, shell removal is undesirable, as the process destroys the shell and prevents further analysis, deforms tissue, and increases the complexity of the sample preparation and final analysis. Moreover, destructive methods are unacceptable for use with rare samples.

Using techniques capable of imaging in high contrast would enable nondestructive imaging, which would not only address the above problems, but also enable new types of studies, e.g., in vivo imaging. Higher contrast could also enable the study of shell features such as growth rings, which, similar to soft tissue, must be studied through destructive methods. Growth rings are a common feature in shells that form due to episodic growth, i.e., growth in bursts. Similar to studying tree rings (dendrochronology), growth rings can provide valuable information about the specimen's age, as well as environmental factors experienced during the course of its life[11]. Shell growth can result in clear external features, such as ridges, but these features might not be reliable for age determination. Internal structures are more reliable, but they cannot be observed in conventional micro-CT imaging[10]. Therefore, mechanical sectioning with diamond cutters and optical microscopy is typically used.

X-ray phase-contrast methods have emerged as powerful techniques to improve the often lacking contrast in conventional attenuation-based X-ray imaging. This improved contrast is obtained by utilizing the wave nature of X-rays. To describe interactions with X-rays, the complex refractive index $n = 1 - \delta + i\beta$ is often used. The two parameters $\delta$ and $\beta$ depend on both energy and atomic composition and correspond to phase shift and absorption, respectively. Absorption is one of the main effects in conventional attenuation-based X-ray imaging. The improvement seen in phase contrast stems from the addition of a new contrast mechanism—phase shift. Phase contrast is also more beneficial for materials of low atomic number (Z) since phase shift is a stronger effect for low Z materials than absorption. For soft biological tissue and other materials consisting mainly of low Z elements, $\delta$ is orders of magnitude larger than $\beta$ in the relevant energy range. This shows the potential to achieve significantly higher sensitivity in imaging. Even for shells, with higher Z, the advantage is substantial. As both attenuation and phase shift are proportional to the material density, the image will be a function of density. Phase contrast can thus be used to capture differences in density more clearly. A key application of X-ray phase contrast imaging has been in soft-tissue imaging in research, but clinical applications have also been explored[12–14]. Propagation-based imaging (PBI) is one of the most widely used techniques to capture phase contrast. This technique relies on spatial coherence which in practice means a source producing near parallel radiation or a small source emission spot[15–17]. This typically results in either large and expensive sources or limitations in power. PBI utilizes both large synchrotron sources and compact sources to improve micro-CT[18,19], but dense structures such as shells remain an issue. For small (3 mm) bivalves, good results have been reported[20,21], but no larger specimens have been imaged. This is most likely due to the negligible soft tissue contrast achieved with attenuation-based contrast when sufficiently high X-ray energy is used to penetrate the shell. Moreover, brain imaging, a conceptually similar problem that involves soft tissue encased in bone, can benefit from phase contrast as shown by Croton et al.[3] Furthermore, because PBI reproduces high spatial frequencies with higher contrast, it can reveal fine details in samples that cannot be imaged with an equivalent attenuation-based setup, even though technically the resolution is the same. This can be used in samples with low atomic numbers such as soft biological tissue, and in calcium-rich structures, such as bones and shells[22,23]. For attenuation-based imaging, the contrast can be adjusted with energy to match either soft tissue or hard tissue. The strong dependence on Z means that the energy cannot be matched to yield high contrast for both tissue types in a single image. For phase contrast, the weaker dependence on Z and boost of contrast, especially in low Z materials, means that soft and hard tissue can be imaged simultaneously. For such demanding tasks the properties of the source are crucial, and cannot always be met on laboratory sources.

Here we show how PBI significantly improves tissue contrast without shell removal in six bivalves and gastropods. Moreover, PBI reveals features in the shell that were previously invisible in micro-CT images, such as growth lines, and can differentiate between calcite and aragonite crystal forms.

## Results

Six species from the two most numerous classes of mollusks, gastropods and bivalves, were collected in Japan (see Fig. 1) and imaged in a synchrotron X-ray setup (see Fig. 2). Several specimens of each species were imaged to identify representative features. For each species, four types of images were acquired: images with/without phase contrast and images with/without shell decalcification, i.e., shell removal by dissolving the shell in acid (see Supplementary Fig. 1).

Phase-contrast imaging significantly increased the contrast of anatomical features in all cases. The differences are described in more detail below after the types of features studied, including soft tissue, growth rings, and shell crystal forms.

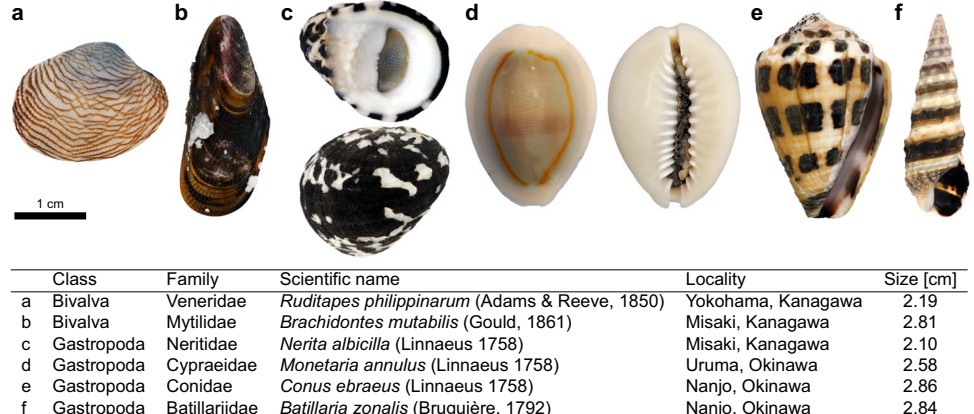

| | Class | Family | Scientific name | Locality | Size [cm] |
|---|---|---|---|---|---|
| a | Bivalva | Veneridae | *Ruditapes philippinarum* (Adams & Reeve, 1850) | Yokohama, Kanagawa | 2.19 |
| b | Bivalva | Mytilidae | *Brachidontes mutabilis* (Gould, 1861) | Misaki, Kanagawa | 2.81 |
| c | Gastropoda | Neritidae | *Nerita albicilla* (Linnaeus 1758) | Misaki, Kanagawa | 2.10 |
| d | Gastropoda | Cypraeidae | *Monetaria annulus* (Linnaeus 1758) | Uruma, Okinawa | 2.58 |
| e | Gastropoda | Conidae | *Conus ebraeus* (Linnaeus 1758) | Nanjo, Okinawa | 2.86 |
| f | Gastropoda | Batillariidae | *Batillaria zonalis* (Bruguière, 1792) | Nanjo, Okinawa | 2.84 |

**Fig. 1 Overview of species imaged. a–f** Photographs of the six mollusk species studied. Locality indicates where the specimens were collected. The size is the longest dimension of the shell.

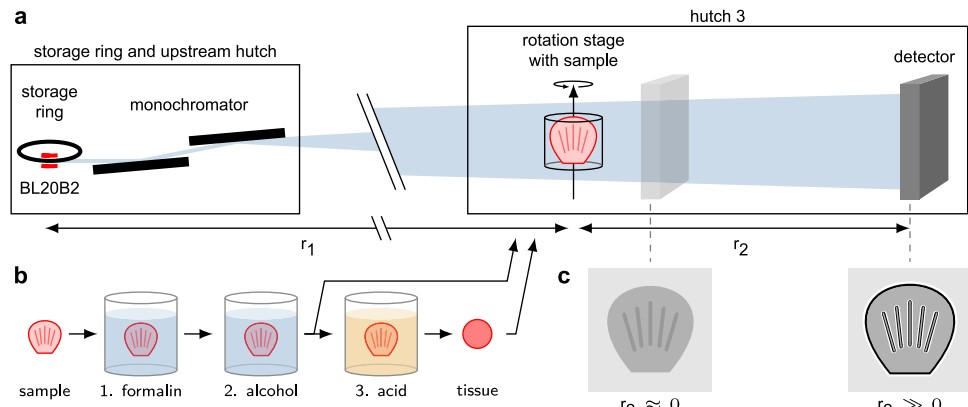

**Fig. 2 Phase-contrast imaging setup (not to scale). a** The SPring-8 BL20B2 bending magnet produces a beam that is filtered with a double-crystal monochromator. After a long distance in vacuum (not shown) the beam enters hutch 3, where the sample and detector are arranged with a variable sample-detector distance $r_2$. The samples are mounted on a rotation stage. The detector is angled with two lens systems and a sensor mounted perpendicular to the X-ray beam direction (not shown). **b** The fresh samples are processed with two or three treatment steps depending on whether the shells are removed. Following collection, all samples are fixed in formalin (1) and stored in alcohol (2). The last step (3) consists of decalcification with acid. The samples are treated with or without step 3, resulting in two sets of samples. **c** The impact of the propagation distance. With small $r_2$ phase effects are negligible, but for large values, significant enhancement occurs.

**Soft-tissue contrast**. When the soft-tissue contrast in phase contrast and conventional attenuation-based contrast imaging is compared, a clear pattern emerges. Phase contrast imaging significantly enhances soft-tissue contrast in images with and without the shell (see Supplementary Fig. 1). The same X-ray energy (40 keV) was used in all imaging to keep as many parameters as possible the same to simplify the comparison between images. A lower energy would, however, be more optimal for the decalcified samples. A direct comparison between calcified and decalcified images should not be made, only between with or without phase contrast. The digestive system and organs such as gonads and ctenidium are largely invisible in attenuation-based imaging; however, these structures can be readily observed in great detail in PBI (see Fig. 3). This is even true for denser structures, such as odontophoral cartilage and the radula (see Fig. 4). Some regions in the tissue remain largely unchanged, most likely due to small variations in the density. When the soft tissue is not obstructed by the shell due to decalcification, the contrast is improved further in some regions; however, this comes at the cost of tissue deformation and the loss of the shell.

**Growth rings**. In contrast to conventional imaging, phase-contrast imaging can be used to visualize growth rings in many

cases (Fig. 5). This is shown for both gastropods (*Monetaria annulus* and *Batillaria zonalis*) and bivalves (*Ruditapes philippinarum*). Some growth lines are difficult to capture and are visible only as slight grayscale shifts, while others are more easily identified. The former situation is observed with *Monetaria annulus* with growth lines appearing as thin curved white lines with a spacing that varies between 0.04 mm and 0.08 mm. In contrast, *Ruditapes philippinarum* shows the opposite situation, with clear growth lines in the outer layer of the shell. Nevertheless, these features are not visible in conventional micro-CT.

**Shell crystal forms**. The main component in mollusk shells is calcium carbonate ($CaCO_3$). It appears in two crystal forms (polymorphs), aragonite and calcite. Aragonite and calcite have densities of $2.93\ \text{g/cm}^3$ and $2.71\ \text{g/cm}^3$, respectively[24]. Phase-contrast imaging has higher sensitivity to density variations than attenuation-based imaging; thus, in species exhibiting both forms of $CaCO_3$, the spatial distribution and volume can be measured with phase-contrast imaging. The ratio of crystal forms is affected by environmental factors and can thus be of interest in ecological studies. *Nerita albicilla* is an example of a species that has layers with both types of $CaCO_3$[25]. The outer layer is calcite, and the inner layer is aragonite. Aragonite absorbs slightly more X-rays

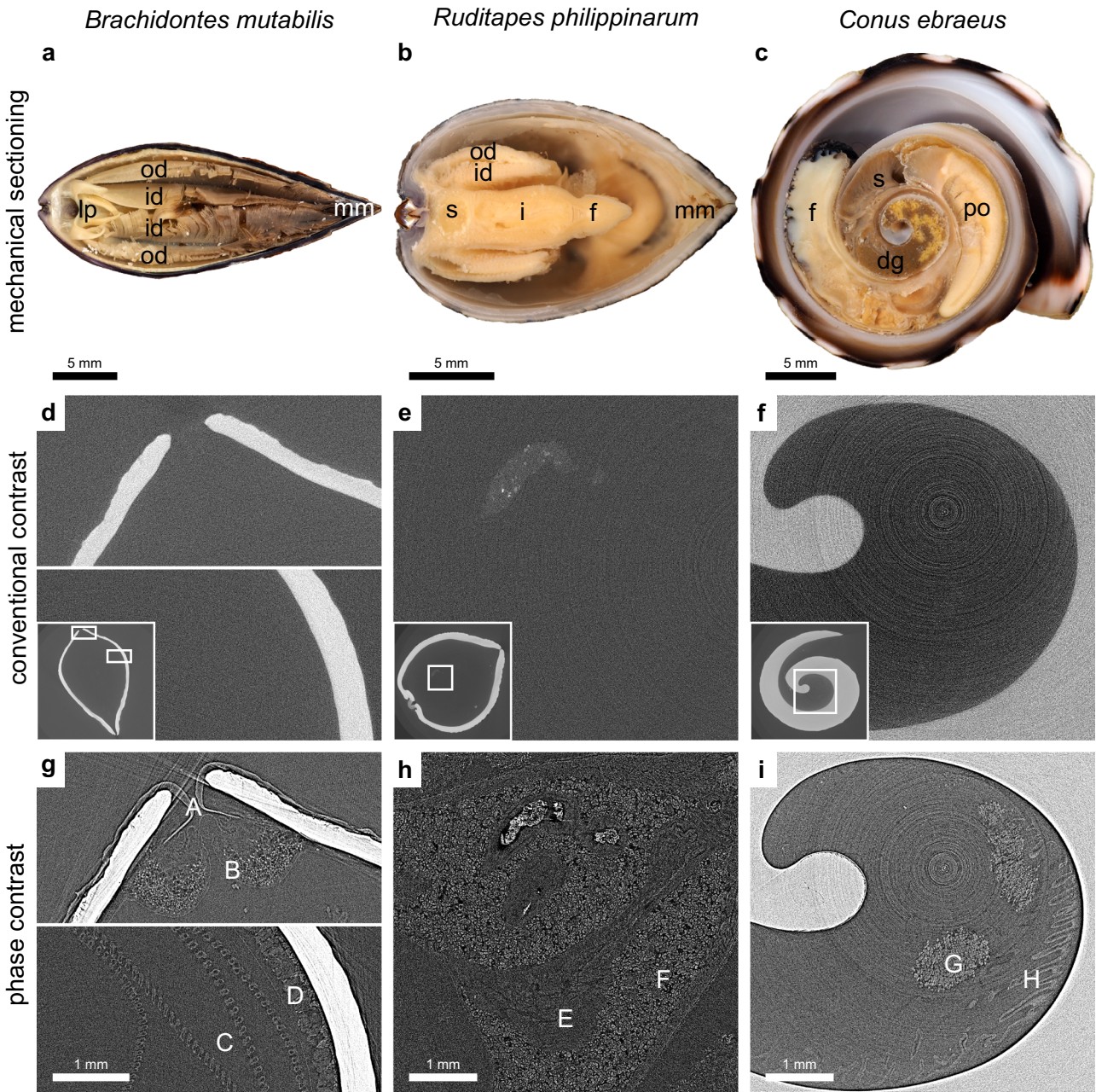

**Fig. 3 Phase contrast enables a drastic improvement in soft-tissue visualization.** To give a general understanding of the soft-tissue anatomy, the upper row shows mechanical sectioning of three species: **a** *Brachidontes mutabilis*, where the inner demibranch (id), outer demibranch (od), labial pulp (lp), and mantle margin (mm) are marked; **b** *Ruditapes philippinarum*, where the foot (f), intestine (i), mantle margin (mm), inner demibranch (id), outer demibranch (od) and stomach (s) are marked; and **c** *Conus ebraeus* where the foot (f), digestive gland (dg), pallial oviduct (po), and stomach (s) are marked. The second and third rows show the corresponding CT slices for conventional contrast and phase-contrast imaging. In (**g**) PBI reveals a large number of structures, including the periostracum (A), mantle margin (B), ctenidium (C), and gonad (D) which do not appear in (**d**). The same is true for (**e**, **h**) where new features appear, such as the intestine (E) and digestive gland (F). In (**i**) the contrast is still lacking, but there is still a clear improvement compared with (**f**), which allows the hypobranchial gland (G) and ctenidium (H) to be observed in the right part of the image. The scales in the corresponding images in the second and third rows are the same.

than calcite and is thus brighter in reconstructed images (Fig. 6). However, this difference is essentially negligible when compared with the noise level in conventional imaging. Thus, separating the two layers based on the small difference in the intensity is very difficult. However, in PBI, the two layers are visually separable due to the edge enhancement inherent to PBI. If phase retrieval is applied, the edge enhancement becomes an intensity difference, which simplifies automatic segmentation which is beneficial for further visualization and quantification.

## Discussion

Phase-contrast imaging can be used to visualize structures in shells and the soft tissue encased therein, which is not possible with conventional micro-CT. There are several clear benefits of using the proposed method. This approach enables imaging of samples that are too valuable to destroy with conventional destructive imaging methods and accelerates the imaging process by removing sample preparation steps and postprocessing. This reduced time per specimen can enable more comprehensive

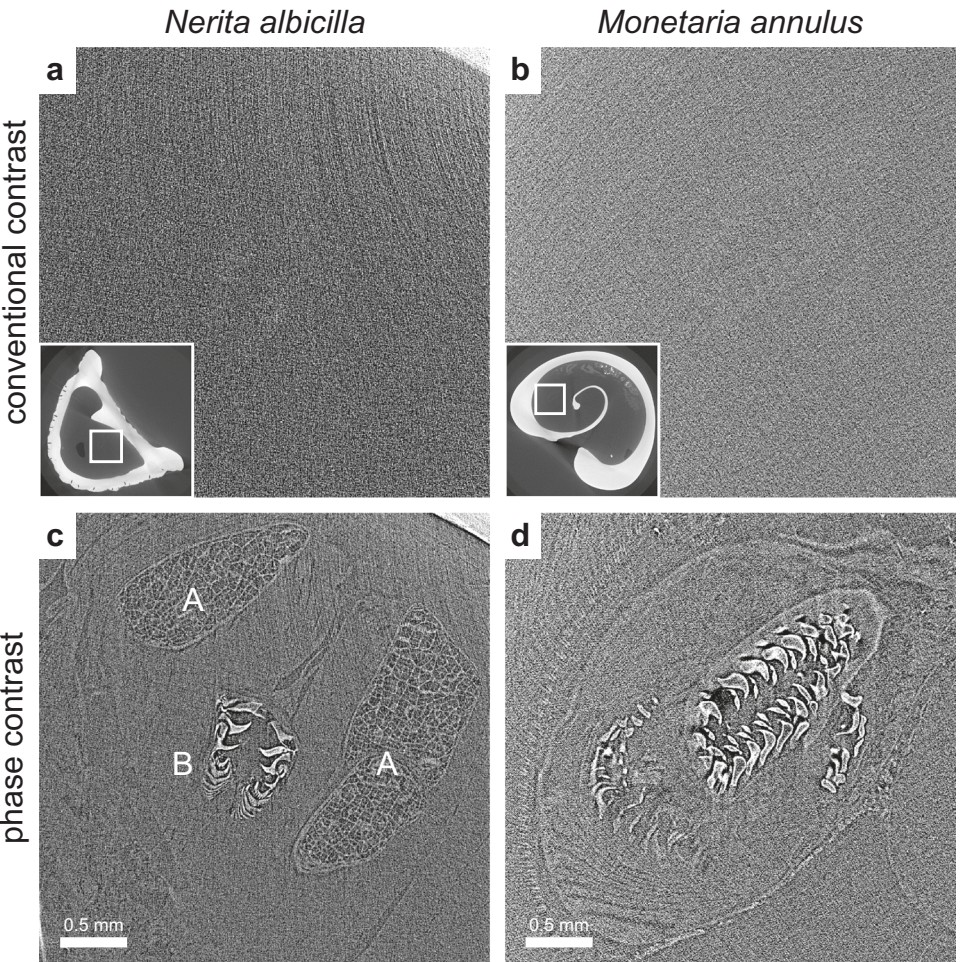

**Fig. 4 Difference in contrast for the radula in conventional attenuation-based contrast and phase contrast imaging.** In (**a**, **b**) which show attenuation-based contrast images, the structures are practically invisible, but with phase contrast, the radular apparatus is shown in great detail. In (**c**) the two odontophoral cartilages (A) and the radula itself (B) are visible, and in (**d**) the details in the surrounding soft tissue are also shown. The scales in the corresponding images in the first and second rows are the same.

studies, which are especially important in fields such as taxonomy and ecology. Fast nondestructive imaging also enables new types of studies, such as imaging living mollusks in shells; and dynamic imaging, which can capture movements in the soft tissue.

However, the soft-tissue contrast in the images with the shell intact is still lower than that in phase-contrast micro-CT images of soft tissue without the absorbing shell. Without the shell, the tissue can be processed further, e.g., by dehydration and paraffin embedding, which increases the contrast. This inevitably leads to trade-offs. Leaving the shell intact results in a decrease in the possible contrast, but prevents destroying the shell and the inevitable tissue distortions that follow. Similar to soft-tissue contrast, performing mechanical sectioning with a diamond cutter and optical microscopy reveals more structures than phase contrast imaging; however, the specimen is destroyed, and the resolution is anisotropic. Sectioning of mollusks for histology can also be complicated by the presence of hard sedimentary grains trapped inside the animal, e.g., in the digestive tract. These internal sediments are difficult to remove and can easily damage the knife edge, hindering the process of serial sectioning. PBI can in other words not presently replace histology, but serve as a useful alternative when a nondestructive method is necessary.

Species that reuse the inner wall during growth, which is known as resorption, such as *Nerita albicilla* and *Conus ebraeus* have thinner shells than other species, which make them suitable for imaging. Species with very thick shells, such as *Monetaria annulus*, which do not use resorption, have less benefits from phase contrast. Thus, the advantages of phase contrast are reduced in species with thicker shells, but it should be noted that the contrast achieved with conventional imaging is very poor, even for species with very thin shells. Therefore, PBI is advantageous in most cases, although the benefit may be moderate for species with very thick shells.

The results achieved in this study rely on recent improvements in the monochromator in the X-ray setup which result in an order of magnitude higher flux[26]. The high flux enables fast image acquisition which reduces the risk of introducing artifacts due to changes in the sample, a non-trivial problem for soft biological samples. All images were acquired with the same improved monochromator, which further shows the limitations of conventional attenuation-based imaging. It should be noted that the optimal energy varies with specimen and technique. Using one energy for all imaging will in other words create a small disadvantage for certain samples and drawing conclusions from small differences in image quality is difficult. However, for the significant difference in contrast observed in this study, the result is quite clear. Although not a limitation of the method itself, the wider adoption of PBI for intact mollusk imaging is complicated by its dependence on spatial coherence and high X-ray flux. Monochromatic radiation is also desirable to avoid beam-

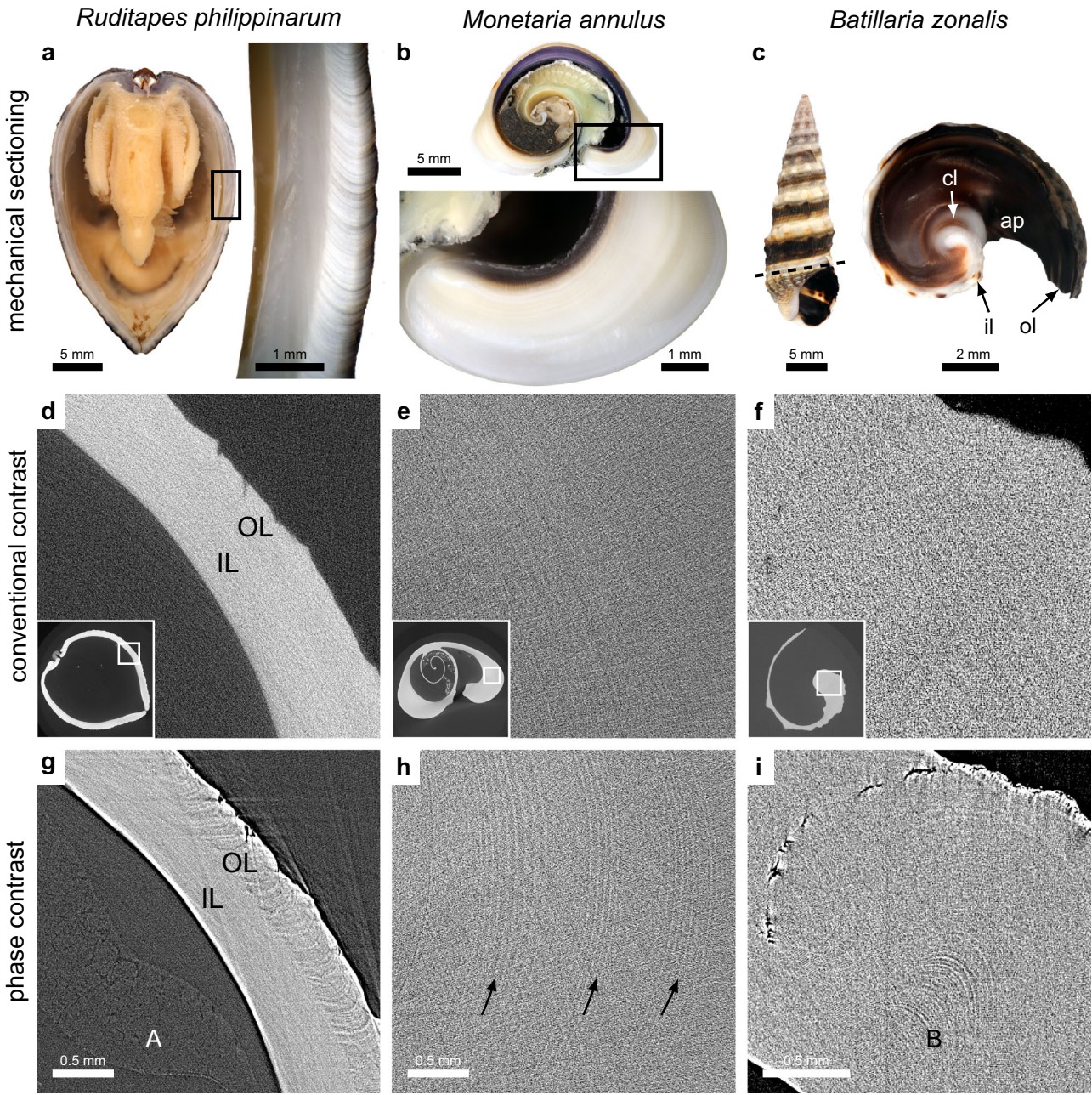

**Fig. 5 Phase contrast enables visualization of growth lines in the shell.** The upper row shows mechanical sectioning of three species: **a** *Ruditapes philippinarum*, **b** *Monetaria annulus*, and **c** *Batillaria zonalis*. In (**c**), the dashed line (left image) shows the sectioning plane. In the right image, four features are marked: the aperture (ap), columella (cl), inner lip (il), and outer lip (ol). The second and third rows show the corresponding CT slices for conventional contrast and phase-contrast imaging. In (**g**) the growth lines in the outer layer (OL) of the shell are visualized by PBI, but not by conventional contrast (**d**). The inner layer (IL) has growth lines in the perpendicular direction (not visible). The mantle tissue is also visible inside the shell (A). In (**h**) groups of thin growth lines are visible inside the shell. Each group of near-vertical white lines is marked with a black arrow. Note that the lines shown in (**e**) are ring artifacts (concentric around the rotation axis of the sample), not actual features. In (**i**) lines are clearly visible in the lower center (B) and the cracks in the upper left region are also better visualized than in (**f**). The scales in the corresponding images in the second and third rows are the same.

hardening artifacts. The flux and brightness of compact laboratory sources are too low to achieve good soft-tissue contrast, even with scan times lasting tens of hours. Thus, synchrotrons are the main tool for performing experiments until brighter compact sources become readily available. Inverse Compton scattering sources are an exception that provides synchrotron-like capabilities suitable for biomedical imaging in a compact format[27], but cost and size are still limiting wider adoption.

Despite the significant improvement in contrast, further contrast enhancements are worth investigating. It is particularly important for studying larger mollusks with very thick shells. The noise and dynamic range of the X-ray detectors are two critical factors for imaging low-contrast features in CT. Although a state-of-the-art sensor was used in this study, the sensor has such a large impact that that future technological developments may be critical in achieving better imaging results. The focus of this work was to evaluate the general benefit of PBI for studying anatomical features in different types of mollusks. For feature detection and studying fine structures, images without phase retrieval may be preferable, which is why most images shown only have basic

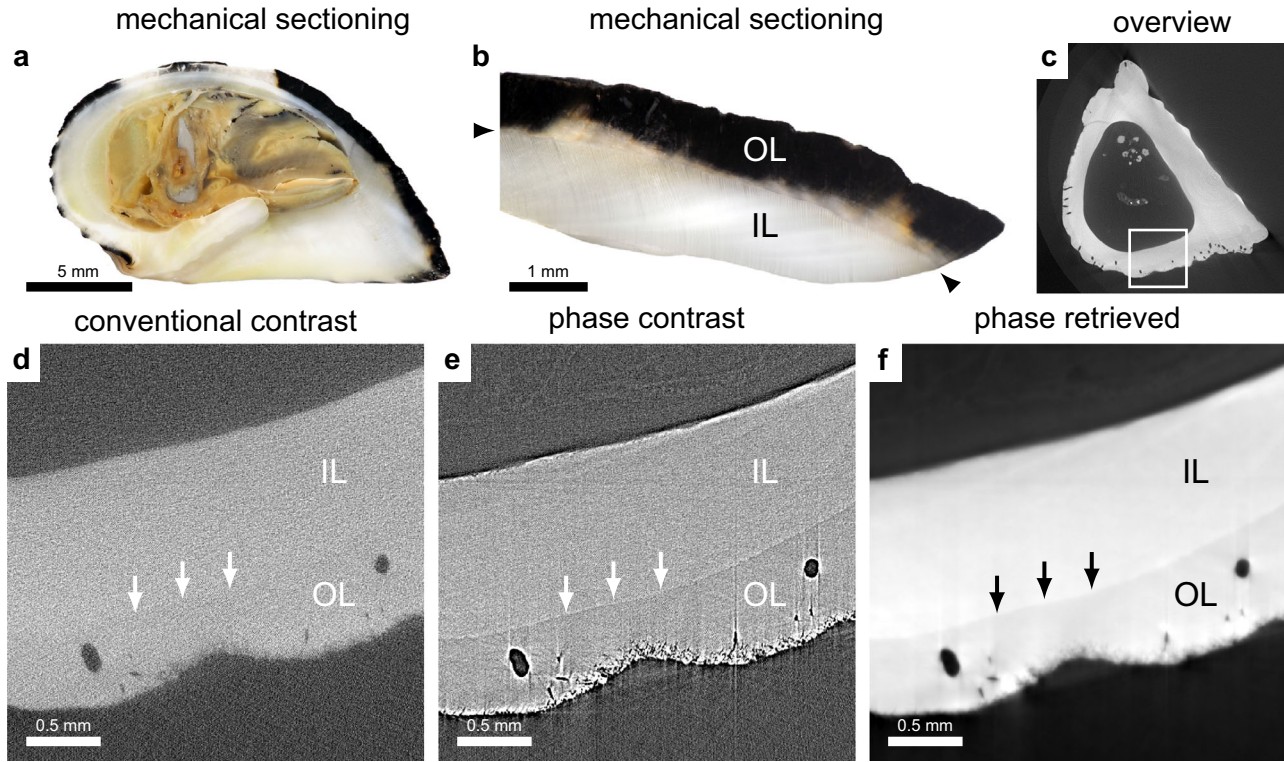

**Fig. 6 Phase contrast enables differentiation between calcium carbonate crystal forms in the shell. a** Mechanical section of *Nerita albicilla* which has layers of aragonite (white) and calcite (dark). **b** Magnified section, with the boundary between the crystal layers marked with two arrowheads. IL denotes the aragonite region (inner layer) and OL denotes the calcite region (outer layer). **c** CT overview. **d** CT slice with conventional contrast. The crystal layer boundary is marked with three arrows. **e** Same slice with phase contrast. **f** Same slice with phase contrast and phase retrieval. The holes in the shell are from Bryozoa.

image corrections (flat-field and dark-field corrections). If segmentation and other tasks are considered, it may be beneficial to apply more involved phase retrieval methods such as multimaterial phase retrieval[28, 29], which could be particularly advantageous if soft tissue and bone are studied simultaneously[30]. Here, the fast Paganin method[31] was used due to the large (multi TB) amount of data. If the objective was to study the soft tissue near the shell, multi-material methods could be worth the additional complexity and computational time.

In conclusion, propagation-based phase contrast imaging can significantly increase soft-tissue contrast in specimens with calcium-rich absorbing exteriors, and the sensitivity of the method allows structures in the exoskeleton, such as growth rings and crystal-form boundaries, to be revealed.

## Methods

**Sample preparation**. Six species of marine mollusks were collected by hand on tidal flats and rocky coasts during April and May at four locations in Kanagawa and Okinawa prefectures in Japan (see Fig. 1). The specimens were stored in a wet environment during collection and transport. Living specimens were killed by submersion in formalin within 8 h. All samples were treated by submersion in a 10% formalin solution and then stored in 70% ethanol. Decalcification of shells was performed with a 10% hydrochloric acid (HCl) solution. Samples were imaged submerged in saltwater in a 20 mm diameter polypropylene (PP) tube.

**Phase-contrast imaging**. Propagation-based phase-contrast imaging was performed at beamline BL20B2[32] at the SPring-8 Synchrotron in Hyogo, Japan (Proposal 2022A1565). A schematic diagram of the setup is shown in Fig. 2. The recently installed double multilayer monochromator (DMM) was used to achieve a quasimonochromatic ($\Delta E/E \sim 4.2\%$) X-ray beam with high flux at 40 keV[26]. A 0.3 mm Cu filter removed any low-energy photons. A rotating piece of bamboo was used as a diffuser to average out fluctuations on a few ms scale. Experiments were performed in hutch 3, resulting in a source-object distance ($r_1$) of ~207 m. The object-detector distance ($r_2$) was varied between 0 m and 1.3 m. The propagation distance ($z$), which is equal to $r_2$, was chosen based on a criterion that achieve the best contrast at a certain frequency, $z = 1/(2\lambda f^2)$[16,33]. This formula assumes a phase object but gives a rough estimate of the true value. $1/f$ was set according to the point spread function (PSF) of the detector. The PSF was measured to ~9 μm with a JIMA RT RC-05 test target. The detector consisted of a single crystal scintillator (200 μm GAGG(Ce)), a lens unit, and a Hamamatsu ORCA Lightning sCMOS camera (C14120-20P). The camera had a $4608 \times 2592$ sensor with 5.5 μm pixels and a 16-bit ADC. The lens unit (two $f = 105$ mm lenses) resulted in an effective pixel size of 5.64 μm (at $r_2 = 0$). At 1.3 m, the slight divergence in the X-ray beam resulted in magnification and thus an effective pixel size of 5.59 μm. An exposure time of 40 ms per image was used. Images were acquired with Hamamatsu HiPic and custom software at the BL20B2 beamline. Tomographies were performed over 180° with 0.1° steps resulting in 1800 projections. The total scan time was <3 min. The dose rate was ~3.0 Gy/s. The transmission through the sample holder (tube filled with water) at the thickest point was between 50% and 60%. For decalcified samples the transmission did not change more than a few percent due to the similarity in density between water and tissue, with a few exceptions, such as sedimentary grains that had significantly lower transmission. For samples with intact shells the transmission varied from 60% down to 3%.

**Processing and analysis**. Data processing and analysis was performed with standard methods in MATLAB R2021b and ImageJ 1.53. Images were dark-field and flat-field corrected. A standard filtered back projection (FBP) reconstruction method was used. In addition, $5 \times 5$ binning was applied to the overview images. Grayscale values were adjusted for each image to best show the features of interest. All images were finally converted to 8-bit. The image in Fig. 6f was processed with Paganin phase retrieval[31]. The filter parameter ($\delta/\mu = 1 \cdot 10^{-9}$ m) was adjusted manually to reduce streaks and improve contrast while simultaneously limiting blurring.

**Reporting summary**. Further information on research design is available in the Nature Portfolio Reporting Summary linked to this article.

## Data availability

The tomography data supporting the findings of this study are available from the corresponding author upon reasonable request.

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

## Acknowledgements

This work was funded by the Japan Society for the Promotion of Science (Kakenhi grant number 21F21759) and supported by SPring-8 proposal 2022A1665. The authors would like to thank MSc Yu Maekawa and MSc Taro Yoshimura for assistance with data handling.

## Author contributions

I.H. and T.S. conceived the experiments. I.H. and T.S. collected the specimen. T.S. prepared the specimens. I.H., M.H., K.U. and T.S. conducted the imaging experiments. I.H. and T.S. analysed the results. I.H. wrote the manuscript with input from all authors.

## Competing interests

The authors declare no competing interests.
