## [Peer Review File · Communications Biology]

Reviewers' comments:

Reviewer #1 (Remarks to the Author):

This is an exciting paper showing the huge potential of non-invasive x-ray tomography in revealing microscopic details of the shell and soft tissues in gastropod and bivalve shells. I recommend accepting as is.

Reviewer #2 (Remarks to the Author):

Congratulations to the authors on an excellent paper, it was a pleasure to read and the images are exquisite. All the statements within are well-justified, the methods are clearly explained, the figures are informative, well-chosen and contain all pertinent information. The results suggest great promise for the application of this technique in further studies of mollusks and are clearly a step forward from their previous studies (referenced within).

It is rare that I cannot find anything to suggest, and I think that reflects the high quality of the manuscript contents and presentation. I have only some optional suggestions for additional references.

Where the authors first mention propagation-based phase contrast x-ray imaging, they cite: Wilkins, S. W., Gureyev, T. E., Gao, D., Pogany, A. & Stevenson, A. W. Phase-contrast imaging using polychromatic hard x-rays. *Nature* 384, 335–338, 10.1038/384335a0 (1996).

Given their work is performed at a synchrotron with monochromatic hard x-rays, they could also consider citing:

Cloetens, P., Barrett, R., Baruchel, J., Guigay, J.P. and Schlenker, M., 1996. Phase objects in synchrotron radiation hard x-ray imaging. *Journal of physics D: applied physics*, 29(1), p.133. and

Snigirev, A., Snigireva, I., Kohn, V., Kuznetsov, S. and Schelokov, I., 1995. On the possibilities of x-ray phase contrast microimaging by coherent high-energy synchrotron radiation. *Review of scientific instruments*, 66(12), pp.5486-5492.

When discussing the brightness limitations of existing laboratory sources and the need for synchrotron sources (line 113), the authors could consider referencing relevant work at compact light sources (e.g. Töpperwien, M., Gradl, R., Keppeler, D., Vassholz, M., Meyer, A., Hessler, R., Achterhold, K., Gleich, B., Dierolf, M., Pfeiffer, F. and Moser, T., 2018. Propagation-based phase-contrast x-ray tomography of cochlea using a compact synchrotron source. *Scientific reports*, 8(1), pp.1-12.)

When discussing multi-material phase retrieval, the authors could consider referencing:

Beltran, M.A., Paganin, D.M., Uesugi, K. and Kitchen, M.J., 2010. 2D and 3D X-ray phase retrieval of multi-material objects using a single defocus distance. *Optics Express*, 18(7), pp.6423-6436.

Reviewer #3 (Remarks to the Author):

The manuscript shows that propagation-based phase-contrast X-ray imaging reveals soft tissues inside mollusks with intact shells. The authors present this nondestructive technique as an alternative to destructive methods, such as histology using mechanical sectioning combined with light microscopy, for obtaining accurate 3D anatomy of mollusks. However, the authors failed to show this. There was no direct comparison made to show that phase-contrast x-ray imaging could rival that of conventional histology. Although images of mechanically sectioned samples were shown, the scales are very different and do not show what is the state-of-the-art in histology of mollusks. Can the scientists in the field of malacology really forget about sectioning their samples and opt for phase-

contrast x-ray CT for their research? A convincing answer should be provided. Instead, comparisons between attenuation-contrast and phase-contrast x-ray imaging were made. But the conclusion here is not new. Phase-contrast x-ray imaging has been shown thousands of times to be superior against attenuation-contrast imaging especially when differentiating between soft-tissues. The main challenge that was addressed by the work was the difficulty of obtaining phase-contrast x-ray CT images of mollusks. But this is a technical advance. The authors themselves highlighted that the results rely on recent improvements in the monochromator which result in higher flux. Unfortunately, I cannot recommend this work for publication as it is.

Reviewer #4 (Remarks to the Author):

The authors quite impressively show the improved visualization of specimens containing high and low absorbing materials using PBI compared to attenuation contrast. The topic is not only relevant to the study of mollusks, similar specimen systems are for example also common in biomedical research.

Comparing CT data from different modalities is always a challenging task, because so many parameters have an effect on the data, including the choice of scan parameters and, prominently, post-processing. I can therefore agree with the choice of the authors to give a qualitative comparison of the result from the two contrasts, without any quantification.

Since comparison is so challenging, I would recommend alleviating or rephrasing some statements in the manuscript, list below.

One thing that needs including is some quantification of the transmission or attenuation values of the shells compared to the soft tissue. Potential readers with similar specimen systems (e.g. wanting to measure brains inside the skull, as mentioned by the authors in the introduction) might greatly benefit from this information when choosing parameters for their own experiments.

Line 8: "Features that are completely invisible in conventional attenuation..." How much effort was put into visualizing these features in the attenuation data? I recommend weakening this statement, maybe something along the lines of adding "under the same scan conditions".

Line 17: "virtual histology" is not equivalent to CT, rather a sub-field or even a catch phrase. I am unsure if it is appropriate in the field of malacology (not my expertise)

Lines 17-19 "The drawback of this method..." In application where CT is used for virtual histology, phase contrast is quite often, if not routinely, used. Contrast in soft tissue, especially if somehow prepared, e.g. embedded in paraffin wax, is quite good and comparable with contrast found in stained histological sections.

Lines 31 to 33 "Improving contrast would enable..." I suggest reformulating this sentence, as it sounds like improving contrast has not been attempted or done before.

Line 39-40 I suggest adding a more detailed explanation of why phase contrast is suited, especially in low Z materials or specimens containing both low and high Z materials.

Line 42 "This technique relies..." I recommend rephrasing with coherence in mind, as not all readers might be familiar with the X-ray concept of brightness.

Line 44 "Virtual histology" I find the interchanging use of virtual histology and CT confusing. I suggest sticking to CT in this context, especially since the data in this study are not really compared to actual histology.

Line 45 "no larger specimens have been imaged" Why are larger specimens a challenge?

Line 46 "Because PBI is sensitive to high spatial frequencies" I can get on board with this statement, but it needs more detail and rephrasing. PBI is sensitive to the second derivative of the phase shift and can pick up small features even in extended specimens where attenuation might struggle. Is it, however, intrinsically better than attenuation for high resolution?

Line 48-50 As with line 39-40, I recommend explaining the underlying problem, and why phase contrast is suited. Sensitivity with attenuation contrast can be tuned to either hard or soft tissue, not both simultaneously. The authors mention this, but it should be stated more clearly.

Line 63 "...contrast in images with and without shell..." It should be noted that 40 keV is not an optimized energy for attenuation contrast and specimens with soft tissue only (no shell).

Line 78 This needs explaining. Attenuation is directly proportional to mass density, and the phase is directly proportional to electron density, so why the improved sensitivity?

Line 80 "quantitatively" what is meant here? Quantitatively suggest that some physical quantity like electron density is measured, which is not reliable with PB phase contrast.

Line 85 "If phase retrieval is applied..." Paganin phase retrieval is a smoothing filter in projection space. I could imagine that, if a smoothing filter is applied in attenuation, density differences might become visible above the noise level.

Figure 6: The last point of the caption should be "f", not "e"

Line 108-109 A higher flux surely results in faster acquisition speeds, but does it improve data quality?

Line 109 The fact that the data were acquired on the same instrument is a plus for the comparison. However, the chosen energy might not be optimal for attenuation imaging.

Line 111 PB phase doesn't have strict requirements on monochromaticity and can be used with pink beam, and even in lab sources edge enhancement can be observed

Line 115 and following: I completely agree with the authors. I would love to see a more in-depth analysis of the data in the recent future.

Dear Editor and Reviewers,

We thank you for providing constructive feedback and thoughtful comments, which we used to improve our manuscript. Below is our point-by-point response (blue) including additions/changes to the revised manuscript (red) according to the reviewer comments (black). Line numbers referred to in this response relate to the new revised manuscript.

Yours sincerely,

The authors

Reviewer #1

This is an exciting paper showing the huge potential of non-invasive x-ray tomography in revealing microscopic details of the shell and soft tissues in gastropod and bivalve shells. I recommend accepting as is.

Response: We thank the reviewer for the positive recommendation.

Reviewer #2

Congratulations to the authors on an excellent paper, it was a pleasure to read and the images are exquisite. All the statements within are well-justified, the methods are clearly explained, the figures are informative, well-chosen and contain all pertinent information. The results suggest great promise for the application of this technique in further studies of mollusks and are clearly a step forward from their previous studies (referenced within). It is rare that I cannot find anything to suggest, and I think that reflects the high quality of the manuscript contents and presentation. I have only some optional suggestions for additional references.

Response: We thank the reviewer for the strong recommendation.

Where the authors first mention propagation-based phase contrast x-ray imaging, they cite:

Wilkins, S. W., Gureyev, T. E., Gao, D., Pogany, A. & Stevenson, A. W. Phase-contrast imaging using polychromatic hard x-rays. *Nature* 384, 335–338, 10.1038/384335a0 (1996).

Given their work is performed at a synchrotron with monochromatic hard x-rays, they could also consider citing:

Cloetens, P., Barrett, R., Baruchel, J., Guigay, J.P. and Schlenker, M., 1996. Phase objects in synchrotron radiation hard x-ray imaging. *Journal of physics D: applied physics*, 29(1), p.133.

and

Snigirev, A., Snigireva, I., Kohn, V., Kuznetsov, S. and Schelokov, I., 1995. On the possibilities of x - ray phase contrast microimaging by coherent high - energy synchrotron radiation. Review of scientific instruments, 66(12), pp.5486-5492.

When discussing the brightness limitations of existing laboratory sources and the need for synchrotron sources (line 113), the authors could consider referencing relevant work at compact light sources (e.g. Töpperwien, M., Gradl, R., Keppeler, D., Vassholz, M., Meyer, A., Hessler, R., Achterhold, K., Gleich, B., Dierolf, M., Pfeiffer, F. and Moser, T., 2018. Propagation-based phase-contrast x-ray tomography of cochlea using a compact synchrotron source. Scientific reports, 8(1), pp.1-12.)

When discussing multi-material phase retrieval, the authors could consider referencing:

Beltran, M.A., Paganin, D.M., Uesugi, K. and Kitchen, M.J., 2010. 2D and 3D X-ray phase retrieval of multi-material objects using a single defocus distance. Optics Express, 18(7), pp.6423-6436.

Response: We have added the suggested references. We also agree that inverse Compton scattering sources are an interesting alternative, but believe it is important to note the limited adoption of these sources.

Line 53: reference 15 and 16

Line 141: "Inverse Compton scattering sources are an exception that provides synchrotron-like capabilities suitable for biomedical imaging in a compact format,²⁷ but cost and size are still limiting wider adoption."

Line 151: reference 28

Reviewer #3

The manuscript shows that propagation-based phase-contrast X-ray imaging reveals soft tissues inside mollusks with intact shells. The authors present this nondestructive technique as an alternative to destructive methods, such as histology using mechanical sectioning combined with light microscopy, for obtaining accurate 3D anatomy of mollusks. However, the authors failed to show this. There was no direct comparison made to show that phase-contrast x-ray imaging could rival that of conventional histology. Although images of mechanically sectioned samples were shown, the scales are very different and do not show what is the state-of-the-art in histology of mollusks. Can the scientists in the field of malacology really forget about sectioning their samples and opt for phase-contrast x-ray CT for their research? A convincing answer should be provided.

Response: While it is true that we present PBI as a nondestructive alternative to histology, we do not claim that PBI is superior to histology or that it can replace it in all situations. We try to make the case that when histology is not an option, for reasons mentioned in the

introduction of the manuscript, recent advances in technology has made PBI a compelling alternative that far surpasses the limitations of conventional attenuation-based imaging.

In traditional histological examinations, the cutting of intact calcareous shells is not possible, necessitating the decalcification of shells. However, when dealing with indestructible specimens, such as type specimens crucial for taxonomy, nondestructive examination as demonstrated in this study remains the only viable option. This underscores the significant importance of the methodology proposed in this paper.

Another issue with histological sectioning is the presence of sediment grains trapped inside animals. The digestive tracts often contain hard sediment grains in mollusks, especially in grazers and deposit feeders. These internal sediments are difficult to remove and can easily damage the knife edge, hindering the process of serial sectioning. In this case, x-ray CT is the best solution to internal examination.

The images of mechanically sectioned mollusks within the manuscript serve primarily to provide readers with a general understanding of the anatomy. To enhance clarity, we have made the following changes.

Line 8: ... aragonite crystal forms. Phase-contrast imaging can thus serve as a compelling alternative when destructive methods are not an option.

Line 120: Sectioning of mollusks for histology can also be complicated by the presence of hard sedimentary grains trapped inside the animal, e.g., in the digestive tract. These internal sediments are difficult to remove and can easily damage the knife edge, hindering the process of serial sectioning. PBI can in other words not presently replace histology, but serve as a useful alternative when a nondestructive method is necessary.

Fig 3: "To give a general understanding of the soft-tissue anatomy, the upper row shows mechanical sectioning of three ...

Instead, comparisons between attenuation-contrast and phase-contrast x-ray imaging were made. But the conclusion here is not new. Phase-contrast x-ray imaging has been shown thousands of times to be superior against attenuation-contrast imaging especially when differentiating between soft-tissues.

Response: It is true that showing the general benefit of phase contrast compared to attenuation-based imaging is not new, as evident of the several references in the manuscript. We believe, however, that showing the benefit of particular applications can be of great interest, especially since the considered application has been a challenging one.

The main challenge that was addressed by the work was the difficulty of obtaining phase-contrast x-ray CT images of mollusks. But this is a technical advance. The authors themselves highlighted that the results rely on recent improvements in the monochromator

which result in higher flux. Unfortunately, I cannot recommend this work for publication as it is.

Response: This final comment that this work relies on a technical advancement is again true, but we don't think that the result is less interesting or important to communicate for that reason.

Reviewer #4

The authors quite impressively show the improved visualization of specimens containing high and low absorbing materials using PBI compared to attenuation contrast. The topic is not only relevant to the study of mollusks, similar specimen systems are for example also common in biomedical research.

Response: We thank the reviewer for the positive recommendation.

Comparing CT data from different modalities is always a challenging task, because so many parameters have an effect on the data, including the choice of scan parameters and, prominently, post-processing. I can therefore agree with the choice of the authors to give a qualitative comparison of the result from the two contrasts, without any quantification.

Since comparison is so challenging, I would recommend alleviating or rephrasing some statements in the manuscript, list below.

One thing that needs including is some quantification of the transmission or attenuation values of the shells compared to the soft tissue. Potential readers with similar specimen systems (e.g. wanting to measure brains inside the skull, as mentioned by the authors in the introduction) might greatly benefit from this information when choosing parameters for their own experiments.

Response: We agree that transmission values could be of interest and have added some values to the method section. The transmission varies between 60% and 4% for samples with shells. This indicates a good match with the energy. For decalcified samples the transmission does not drop much below 60% this is clearly not optimal for best contrast, but it is important to note that the key images for this study are those of samples with intact shells (all images shown in figure 3, 4, 5, and 6). The question of energy is addressed in a later question.

Line 179: "The transmission through the sample holder (tube filled with water) at the thickest point was between 50% and 60%. For decalcified samples the transmission did not change much due to the similarity in density between water and tissue, with a few exceptions, such as sedimentary grains that had significantly lower transmission. For samples with intact shells the transmission varied from 60% down to 3%."

Line 8: “Features that are completely invisible in conventional attenuation...” How much effort was put into visualizing these features in the attenuation data? I recommend weakening this statement, maybe something along the lines of adding “under the same scan conditions”.

Response: We mainly adjusted greyscale and applied phase retrieval, but many features easily seen in PBI were completely drowned in noise in conventional imaging. However, we agree that scan conditions are crucial.

Line 8: “Features that are almost invisible in conventional attenuation-based micro-computed tomography (micro-CT) are clearly reproduced with phase-contrast imaging under the same scan conditions.”

Line 17: “virtual histology” is not equivalent to CT, rather a sub-field or even a catch phrase. I am unsure if it is appropriate in the field of malacology (not my expertise)

Response: Thanks, the mixed used was confusing.

Line 17: “is virtual histology, i.e., micro-computed tomography (micro-CT)”

-> “is micro-computed tomography (micro-CT)”

Lines 17-19 “The drawback of this method...” In application where CT is used for virtual histology, phase contrast is quite often, if not routinely, used. Contrast in soft tissue, especially if somehow prepared, e.g. embedded in paraffin wax, is quite good and comparable with contrast found in stained histological sections.

Response: Yes, we agree. The point was that nondestructive imaging, or rather imaging raw, untreated samples, with micro-CT has lower contrast since it for example can't use paraffin embedding etc., but this was admittedly not so clear. As for phase contrast, we will get to that later in the introduction.

Line 17: “The drawback of this methods is that the soft-tissue contrast is inferior to that of histology, as the contrast mechanism relies only on X-ray attenuation. Preparation of the sample such as paraffin embedding to improve contrast is routinely used, but the treatment can also alter the sample. The problem with limited contrast...”

Lines 31 to 33 “Improving contrast would enable...” I suggest reformulating this sentence, as it sounds like improving contrast has not been attempted or done before.

Response: Yes, this could be confusing. We have modified the sentence.

Line 32: “Using techniques capable of imaging in high contrast would enable nondestructive imaging, which would not only address the above problems, but also enable new types of studies, e.g., in vivo imaging. Higher contrast...”

Line 39-40 I suggest adding a more detailed explanation of why phase contrast is suited, especially in low Z materials or specimens containing both low and high Z materials.

Response: We've tried to keep the physics part short since the main audience is biologists, but we agree that this is an important point to convey.

Line 41: “To describe interactions with X-rays, the complex refractive index $1-\delta+i\beta$ is often used. The two parameters δ and β depend on both energy and atomic composition and correspond to phase shift and absorption, respectively. Absorption is one of the main effects in conventional attenuation-based X-ray imaging. The improvement seen in phase contrast stems from the addition of a new contrast mechanism—phase shift. Phase contrast is also more beneficial for materials of low atomic number (Z) since phase shift is a stronger effect for low Z materials than absorption. For soft biological tissue and other materials consisting mainly of low Z elements, δ is orders of magnitude larger than β in the relevant energy range. This shows the potential to achieve significantly higher sensitivity in imaging. Even for shells, with higher Z, the advantage is substantial.”

Line 42 “This technique relies...” I recommend rephrasing with coherence in mind, as not all readers might be familiar with the X-ray concept of brightness.

Response: We agree that some further explanation is warranted.

Line 52: “This technique relies on spatial coherence which in practice means a source producing near parallel radiation or a small source emission spot. This typically results in either large and expensive sources or limitations in power.”

Line 44 “Virtual histology” I find the interchanging use of virtual histology and CT confusing. I suggest sticking to CT in this context, especially since the data in this study are not really compared to actual histology.

Response: We agree that the terminology was inconsistent. We have done the suggested change.

Line 54: “virtual histology” -> “micro-CT”

Line 45 “no larger specimens have been imaged” Why are larger specimens a challenge?

Response: We can only speculate why no one has done this before, but most likely it is due to the difficulty to get good contrast with most X-ray sources, e.g. due to beam hardening. Larger specimens will also require higher energy where X-ray interaction and thus contrast is even lower. This is, however, mentioned in the first paragraph of the introduction

Line 56: “This is most likely due to the negligible soft tissue contrast achieved with attenuation-based contrast when sufficiently high X-ray energy is used to penetrate the shell.”

Line 46 “Because PBI is sensitive to high spatial frequencies” I can get on board with this statement, but it needs more detail and rephrasing. PBI is sensitive to the second derivative of the phase shift and can pick up small features even in extended specimens where attenuation might struggle. Is it, however, intrinsically better than attenuation for high resolution?

Response: Well, resolution goes hand in hand with contrast, so it’s hard to speak about one without the other. PBI does not fundamentally change the resolution of the imaging system, but since the contrast transfer function is higher for high spatial frequencies the observable resolution is better. We assume this is what you refer to by “intrinsically better”.

Line 59: “because PBI reproduces high spatial frequencies with higher contrast, it can reveal fine details in samples that cannot be imaged with an equivalent attenuation-based setup, even though technically the resolution is the same. This can be used in samples with ...”

Line 48-50 As with line 39-40, I recommend explaining the underlying problem, and why phase contrast is suited. Sensitivity with attenuation contrast can be tuned to either hard or soft tissue, not both simultaneously. The authors mention this, but it should be stated more clearly.

Response: We modified the end of the paragraph

Line 61: “...bones and shells.^{22,23} For attenuation-based imaging, the contrast can be adjusted with energy to match either soft tissue or hard tissue. The strong dependence on Z means that the energy cannot be matched to yield high contrast for both tissue types in a single image. For phase contrast, the weaker dependence on Z and boost of contrast, especially in low Z materials, means that soft and hard tissue can be imaged simultaneously. For such demanding tasks the properties of the source are crucial, and cannot always be met on laboratory sources.

Line 63 "...contrast in images with and without shell..." It should be noted that 40 keV is not an optimized energy for attenuation contrast and specimens with soft tissue only (no shell).

Response: Yes, 40 keV is not optimal for soft tissue only. In the figure caption (Supplementary Figure 1.) we wrote "Furthermore, the X-ray energy is furthermore optimized for the images in the right column.", i.e. referring to the samples with intact shells. But this deserves to be mentioned in the main text as well.

Supplementary Figure 1.: Furthermore, the X-ray energy is ~~furthermore~~ optimized ...

Line 80: "The same X-ray energy (40 keV) was used in all imaging to keep as many parameters as possible the same to simplify the comparison between images. A lower energy would, however, be more optimal for the decalcified samples. A direct comparison between calcified and decalcified images should not be made, only between with or without phase contrast."

Line 78 This needs explaining. Attenuation is directly proportional to mass density, and the phase is directly proportional to electron density, so why the improved sensitivity?

Response: Electron density is proportional to mass density so both parameters are proportional to density, but the material parameter for phase is considerably larger than attenuation (for this material and energy). This should give a sensitivity that is much higher theoretically. In practice the two parameters are measured in different ways so it's not as simple, but with the excellent X-ray beam at BL20B2 the benefit of phase contrast is clearly used which results in higher contrast or in other words sensitivity to density.

Even though we believe a discussion on this topic is beyond the scope of this article, we have added the following sentences in the introduction.

Line 49: "As both attenuation and phase shift are proportional to the material density, the image will be a function of density. Phase contrast can thus be used to capture differences in density more clearly."

Line 97: "Phase-contrast imaging has higher sensitivity to density variations than attenuation-based imaging"

Line 80 "quantitatively" what is meant here? Quantitatively suggest that some physical quantity like electron density is measured, which is not reliable with PB phase contrast.

Response: By quantitatively we mean that for example the volume of each form can be measured well. It is true that the current sentence could be interpreted as that material parameters are measured. This is of course not the case. We are well aware of the limitations of PBI. We have modified the sentence as follows to make this clearer:

Line 99: “the spatial distribution and volume can be measured”

Line 85 “If phase retrieval is applied...” Paganin phase retrieval is a smoothing filter in projection space. I could imagine that, if a smoothing filter is applied in attenuation, density differences might become visible above the noise level.

Response: While this point certainly has some merit, we have not seen much difference when trying this in practice on this data.

Figure 6: The last point of the caption should be “f”, not “e”

Response: Thank you.

Figure 6 caption: “e” -> “f”

Line 108-109 A higher flux surely results in faster acquisition speeds, but does it improve data quality?

Response: In an ideal situation the speed should not matter, but in practice we believe speed is important. In particular, for fresh biological samples. Changes in temperature, condensation, movements etc. can introduce a lot of artifacts that are detrimental to image quality when low contrast features are studied. We have added one sentence to clarify this

Line 132: “The high flux enables fast image acquisition which reduces the risk of introducing artifacts due to changes in the sample, a non-trivial problem for soft biological samples.”

Line 109 The fact that the data were acquired on the same instrument is a plus for the comparison. However, the chosen energy might not be optimal for attenuation imaging.

Response: Choosing between comparing with the same settings and with settings optimized for each sample is not easy. We chose the former as “optimized” has no simple answer. A slightly lower energy could have been better for the attenuation imaging, but perhaps also for the PBI. The choice of energy was in some sense imposed by the system. The DMM has a single energy, 40 keV as opposed to the standard Si monochromator. The benefit is much greater flux.

Line 134: “It should be noted that the optimal energy varies with specimen and technique. Using one energy for all imaging will in other words create a small disadvantage for certain samples and drawing conclusions from small differences in image quality is difficult. However, for the significant difference in contrast observed in this study, the result is quite clear.”

Line 111 PB phase doesn't have strict requirements on monochromaticity and can be used with pink beam, and even in lab sources edge enhancement can be observed

Response: Yes, indeed, monochromaticity is not necessary for PBI and can be done very well with compact lab sources. Our phrasing should be improved here, but the point is that imaging these types of samples (even with the benefit of PBI) with a compact lab source is near impossible due to beam hardening and low flux. We have rephrased the sentence to clarify this.

Line 137: "Although not a limitation of the method itself, the wider adoption of PBI for intact mollusk imaging is complicated by its dependence on spatial coherence and high X-ray flux. Monochromatic radiation is also desirable to avoid beam-hardening artifacts."

Line 115 and following: I completely agree with the authors. I would love to see a more in-depth analysis of the data in the recent future.

Response: Thank you. We hope to continue this work in the future.

REVIEWERS' COMMENTS:

Reviewer #3 (Remarks to the Author):

The authors addressed thoroughly all comments and suggestions raised by reviewers. This study provides excellent use of nondestructive methods for imaging of shell growth and soft tissue.

Reviewer #4 (Remarks to the Author):

The authors addressed all points in clear and exhaustive fashion.